# Potential SARS-CoV-2 infectiousness among asymptomatic healthcare workers

**Ville N. Pimenoff**[1]*, **Miriam Elfström**[2], **Kalle Conneryd Lundgren**[2], **Susanna Klevebro**[3], **Erik Melen**[3], **Joakim Dillner**[1,2]

**1** Department of Laboratory Medicine, Karolinska Institutet, Stockholm, Sweden, **2** Karolinska University Hospital, Stockholm, Sweden, **3** Department of Clinical Science and Education, Södersjukhuset, Karolinska Institutet, Stockholm, Sweden

* ville.pimenoff@ki.se

**Data Availability Statement:** The data constitutes sensitive data about health of human research subjects and thus cannot be directly deposited openly. However, pseudonymised, individual-level

## Abstract

A majority of SARS-CoV-2 infections are transmitted from a minority of infected subjects, some of which may be symptomatic or pre-symptomatic. We aimed to quantify potential infectiousness among asymptomatic healthcare workers (HCWs) in relation to prior or later symptomatic disease. We previously (at the onset of the SARS-CoV-2 epidemic) performed a cohort study of SARS-CoV-2 infections among 27,000 healthcare workers (HCWs) at work in the capital region of Sweden. We performed both SARS-CoV-2 RT-PCR and serology. Furthermore, the cohort was comprehensively followed for sick leave, both before and after sampling. In the present report, we used the cohort database to quantify potential infectiousness among HCWs at work. Those who had sick leave either before or after sampling were classified as post-symptomatic or pre-symptomatic, whereas the virus-positive subjects with no sick leave were considered asymptomatic. About 0.2% (19/9449) of HCW at work were potentially infectious and pre-symptomatic (later had disease) and 0.17% (16/9449) were potentially infectious and asymptomatic (never had sick leave either before nor after sampling). Thus, 33% and 28% of all the 57 potentially infectious subjects were pre-symptomatic or asymptomatic, respectively. When a questionnaire was administered to HCWs with past infection, only 10,5% of HCWs had had no indication at all of having had SARS-CoV-2 infection ("truly asymptomatic"). Our findings provide a unique quantification of the different groups of asymptomatic, potentially infectious HCWs.

## Introduction

Transmission modelling has indicated that most (about 59%) of SARS-CoV-2 transmission occurs from non-symptomatic individuals with pre-symptomatic spread being more important (about 35% of transmissions) than spread from individuals who never develop any symptoms (i.e. asymptomatic) (about 24% of transmissions) [1–3]. The individual contribution to the SARS-CoV-2 transmission is variable between infectious subjects, with about 20% of infected subjects responsible for about 80% of viral transmissions [4, 5]. When screening non-symptomatic individuals, the proportion of positives that are pre-symptomatic or will remain asymptomatic is not known.

data (N = 9449) that allow full replication of the results in this article are freely available from b2share (https://b2share.eudat.eu/). The separate clinical dataset (N = 3981) of HCWs who completed serology testing and a questionnaire about symptoms is also available in aggregated age-group format in the b2share portal. The data files are also available via the following repository links: Individual HCWs cohort data from KI Hospital (N = 9449) https://doi.org/10.6084/m9.figshare.17114138 Additional data from South Hospital in Stockholm (N = 3981) https://doi.org/10.6084/m9.figshare.17117093.

**Funding:** This work was supported by the Karolinska University Hospital (KUH); the County Council of Stockholm; Erling-Persson family foundation; KTH Royal Institute of Technology; Creades and SciLifeLab. We also wish to thank the Science for Life Laboratory for the antibody testing, the National Pandemic Center of Sweden for the PCR testing and the Karolinska University Hospital for important contributions towards the inception of this study. There were no roles for the funders in design or execution of the study, in analysis and interpretation of data or in the decision to submit for publication.

**Competing interests:** The authors have declared that no competing interests exist.

The first wave of the SARS-CoV-2 epidemic affected Sweden in March-June 2020, particularly in the Stockholm region [6]. The Stockholm HCW study of past or present SARS-CoV-2 invited all healthcare workers (HCWs) in the region for PCR and antibody testing in April-June 2020, with about 27,000 HCWs enrolled [7–9]. Whereas antibodies and low amounts of virus in PCR (high Ct values) were associated with past disease (post-symptomatic subjects) [7, 8], high amounts of virus in PCR predicted future disease in the next 1–2 weeks [7]. The pattern with no symptoms, high amounts of virus and no antibodies was unfortunately particularly common among HCWs working with home care for older persons [10].

The high-level independent expert investigation has emphasized that a failure to recognize the importance of asymptomatic transmissions was a decisive moment that furthered the spread of the pandemic [11]. The few studies that have quantified the importance of asymptomatic spread have been systematically reviewed [12], but it was clear that more data on this point was needed. We realized that the database of our previously performed Stockholm HCW cohort study provided a unique possibility for quantitative estimation of the proportion of pre-symptomatic and asymptomatic subjects among potentially infectious non-symptomatic HCWs. For this ad hoc study, we analyzed the 9449 HCWs who had complete testing and sick leave data, and in addition, compared with a group of 3981 HCWs who completed a questionnaire on symptoms.

## Material and methods

In the previously described Stockholm HCW cohort, all healthcare providers in the capital region of Sweden were asked about participation in a study of past or present SARS-CoV-2 infections [8, 9]. Among the major healthcare providers who agreed to participate, a very high proportion (>90%) of the HCWs at work were enrolled, following written informed consent [8, 9]. This report focuses on the two largest healthcare providers, Karolinska University Hospital and Stockholm South General Hospital. At Karolinska (Fig 1), there were 9449 employees with complete testing data who were also followed for past or future sick leave [7]. At South Hospital the participants instead completed a questionnaire about symptoms (3981 HCWs) [9]. All HCWs at work were eligible for inclusion (physicians, nurses, assistant nurses,

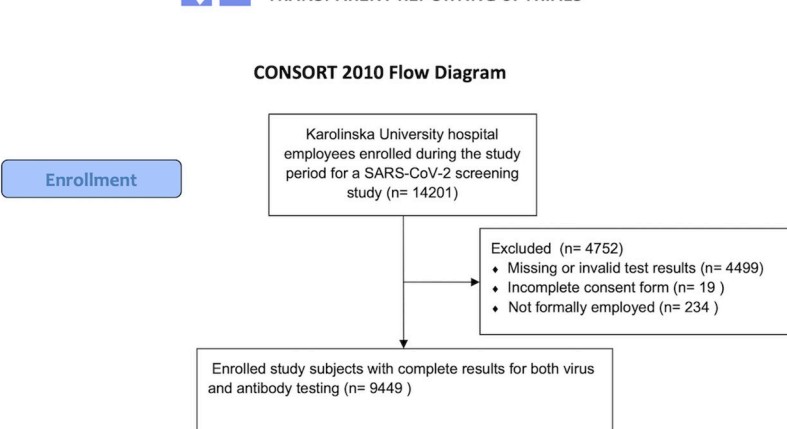

**Fig 1. CONSORT flowchart.** Study flow chart according to the Standard Reporting of Observational Studies (STROBE) guidelines. Abbreviation: SARS-CoV-2, severe acute respiratory syndrome coronavirus 2.

psychologists, social workers, physiotherapists, care administrators, medical secretaries, occupational therapists, speech therapists and nutritionists etc). Rules were strict that HCWs were not allowed to work in case of possible symptoms. We have previously reported that antibodies are strongly associated with past disease but not with future disease [8]. Similarly, we found that presence of low amounts of virus in PCR was associated with past disease, but not with future disease [7]. Conversely, presence of high amounts of virus associated with future disease, but not past disease [7]. For detailed patient flow charts of the cohort study, please see references [7, 9]. In the present report, we considered HCWs who tested positive but did not have the post-symptomatic testing pattern (presence of antibodies or only low amounts of virus) as potentially infectious, as it is well established that subjects are not infectious when returning to work after the stipulated number of days have passed after symptoms have resolved. The study was approved by the Swedish Ethical Review Agency and registered at clinicaltrials.gov (NCT04411576). Informed consents were collected from each participant prior start of the sampling (S1 File).

## Statistical analysis

Proportion and 95% bootstrapped-confidence interval of potentially infectious non-symptomatic HCWs with different clinical characteristics was calculated. The association between age and status as a potentially infectious non-symptomatic HCW was examined using a logistic regression model adjusting for sex and patient contact. Moreover, the association between status as a potentially infectious non-symptomatic HCW and sex, patient contact and sick leave information was independently estimated. All statistical analysis and figure plotting were conducted using either SPSS or R [13].

## Results

The potentially infectious HCWs tended to be more common among younger (<30 years of age) HCWs (OR 2.9 in logistic regression; Table 1). There was a 2.8-fold and 13.5-fold relative risk, respectively, for the potentially infectious subjects of being sick 1–3 weeks before testing (post-symtoomatic) or to become sick in the next two weeks (pre-symptomatic) (Table 2). There were 28.1% (16/57) potentially infectious HCWs who had no sick leave at all (asymptomatic) either before nor after testing (Table 2).

We found no noteworthy differences between men and women or between HCWs with or without patient contact (Table 2). Analysis of the amounts of virus detected in PCR (CT

**Table 1. Proportion of HCWs who are potentially infectious when screening of non-symptomatic HCWs (A) or who have had SARS-CoV-2 infection without experiencing any symptoms (B).**

| A. | Potentially infectious non-symptomatic HCWs | | | | | B. | Seropositives | | Asymptomatic among seropositives | | Total |
|---|---|---|---|---|---|---|---|---|---|---|---|
| Age categories | n | N | % | 95% CI | Adjusted OR (95% CI) | Age categories in years | n | % | n | % | N |
| < 30 years | 12 | 1143 | 1.05 | 0.46–1.64 | 2.9 (1.0–8.4) | <40 | 266 | 20.2 | 32 | 12.0 | 1314 |
| ≥30 to <40 years | 15 | 2347 | 0.64 | 0.32–0.96 | 1.8 (0.6–4.8) | | | | | | |
| ≥40 to <50 years | 17 | 2334 | 0.73 | 0.38–1.07 | 2.0 (0.7–5.5) | ≥40 to <50 | 178 | 18.0 | 17 | 9.6 | 989 |
| ≥50 to <60 years | 8 | 2215 | 0.36 | 0.11–0.61 | 1.0 (0.3–3.1) | ≥50 to <60 | 156 | 16.0 | 12 | 7.7 | 975 |
| ≥60 years | 5 | 1410 | 0.36 | 0.04–0.66 | Ref | ≥60 | 104 | 14.8 | 13 | 12.5 | 703 |
| Total | 57 | 9449 | 0.60 | 0.45–0.76 | | Total | 704 | 17.7 | 74 | 10.5 | 3981 |

n = potentially infectious healthcare workers; N = total of participants.

**Table 2. Clinical characteristics associated to the potentially infectious healthcare workers (n = 57) in a cohort of 9,449 study participants.**

|  | Potentially infectious non-symptomatic | | | | |
|---|---|---|---|---|---|
|  | n (%) | Total % | 95% CI | N | OR (95% CI) |
| **Sex** |  |  |  |  |  |
| Female | 44 (77.2%) | 0.59 | 0.41–0.76 | 7488 | Ref |
| Male | 13 (22.8%) | 0.66 | 0.3–1.02 | 1961 | 1.1 (0.6–2.1) |
| **Patient contact** |  |  |  |  |  |
| No | 15 (26.3%) | 0.49 | 0.24–0.73 | 3073 | Ref |
| Yes | 42 (73.7%) | 0.66 | 0.46–0.86 | 6376 | 1.3 (0.7–2.4) |
| **Sick leave** |  |  |  |  |  |
| No sick leave | 16 (28.1%) | 0.29 | 0.15–0.42 | 5614 | 0.5 (0.2–1.1) |
| 1–2 weeks after testing | 19 (33.3%) | 3.8 | 2.12–5.48 | 500 | 6.9 (3.3–14.6) |
| 1–3 weeks before testing | 11 (19.3%) | 0.81 | 0.33–1.29 | 1356 | 1.2 (0.7–2.2) |
| 4–6 weeks before testing | 11 (19.3%) | 0.56 | 0.23–0.88 | 1979 | Ref |

n = potentially infectious healthcare workers; Total% = proportion from total, 95% CI = 95% Confidence Interval, N = total of participants; Ref = reference group.

values) found a significantly higher amount of virus among the HCWs who were younger than 40-years of age (Fig 2).

Among 3981 HCWs who completed serology testing and a questionnaire about symptoms, 704 subjects tested positive for SARS-CoV-2 antibodies. Among those there were 74 seropositive HCWs (10.5% of seropositives) who responded "No" to the question if they had had any symptoms suggesting that they might have had SARS-CoV-2 infection ("truly asymptomatic"). There was a moderate variability between age groups (ranging from 8–12%), which was not statistically significant (Table 1). Among all HCWs under the age of 40 years, 2.4% were seropositive and asymptomatic.

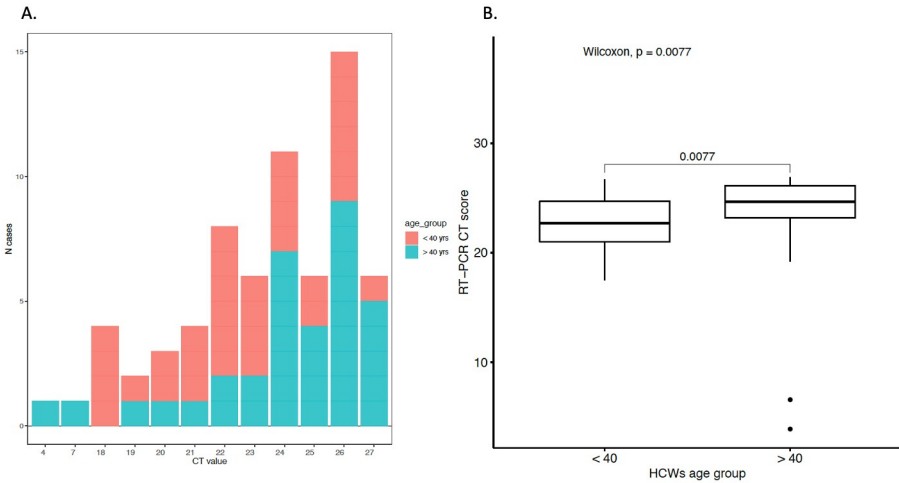

**Fig 2. SARS-CoV-2 RT-PCR CT score distribution among the potential infectious non-symptomatic healthcare workers (HCWs). (A)** Distributions of CT score values among the 57 potentially infectious HCWs. **(B)** Significant difference of the CT score mean between below 40-year-old and 40 or older age potentially infectious non-symptomatic HCWs.

## Discussion

Our study significantly adds up to the topic of what proportion of potentially infectious, non-symptomatic HCWs are either pre-symptomatic, asymptomatic or post-symptomatic. The importance of this topic has been clearly highlighted by the WHO-commissioned independent expert investigation, which highlighted this topic as a decisive factor on the spread of the pandemic [11].

The phenomenon of being non-symptomatic and potentially infectious is more common among younger HCWs, who also are positive for larger amounts of virus, than older HCWs. Although it is significantly more common for these potentially infectious HCWs to be pre-symptomatic (becoming sick in the next few weeks), a meaningful fraction (0.17% of HCWs) had no sick leave at all. That is, when screening non-symptomatic HCWs, about 0.2% (19/9449) were potentially infectious and pre-symptomatic (later had disease) and 0.17% (16/9449) were potentially infectious but never had sick leave neither before nor after sampling corresponding. When a questionnaire was administered to HCWs with past infection, only 10,5% of HCWs had had no indication at all of having had SARS-CoV-2 infection ("truly asymptomatic"). It was strictly forbidden to work in case of symptoms and rules were clear that it was not allowed to just work from home without reporting sick leave in case of symptoms. The lower proportion of subjects being entirely unaware of having had the infection might be due to very minimal symptoms that the subjects did not suspect to be infection at the time they had it.

Modelling studies have indicated that non-symptomatic subjects are responsible for a majority of the transmissions, with transmissions from pre-symptomatic subjects being more important than transmission from subjects who never develop symptoms. However, there is uncertainty in our basic knowledge about how common pre-symptomatic and asymptomatic infection is and our study provides a real-life estimation of this fundamental feature of the infection.

A major strength is that the entire cohort was comprehensively followed with sick leave data, both before and after sampling, a strategy that (as far as we have been able to determine) is internationally unique to our cohort. Other strengths include the fact that we performed both serology and antibody testing and administered questionnaires. By assessing both past and future sick leave we were able to classify the infected HCWs as either pre-symptomatic (coming down with disease in the near future), a-symptomatic (never developing disease) and post-symptomatic (had returned to work after resolution of symptoms). Our study has also some limitations. Firstly, not all HCWs were queried for past symptoms. Secondly, there was no PCR verification of whether sickness was due to SARS-CoV-2 or not, because PCR testing was not generally available at the time.

In this study, we estimate that at a given timepoint during the first wave of COVID-19 outbreak only 57 out of 9449 (0.6%, 95% CI 0.45–0.76) HCWs in Stockholm region were potentially infectious and that only 16/9449 were potentially infectious and remaining healthy (asymptomatic). In addition, our other estimate suggests that only few subjects (10,5%) with past SARS-CoV-2 had never experienced any symptoms[9]. Although the proportion of asymptomatic potential spreaders is low, the fact that they are still at work could result in a relatively large importance for the spread of the epidemic. Compared to the input values assumed in the transmission dynamic modelling, our real-life estimates of how common asymptomatic potentially infectious subjects may be are in good agreement with the input values used in the modelling [1].

Importantly, we find that potentially infectious non-symptomatic HCWs are particularly common among younger HCWs and that the amount of virus also tends to be higher among

younger HCWs. A study serially following young (18–20 years of age) men found a minority of non-symptomatic men who repeatedly tested positive over time [14]. Such long duration and/or serial re-infection without symptoms could be important in the spread of the epidemic.

Taken together, significant SARS-CoV-2 infectivity prior to the onset of symptoms (i.e. pre-symptomatic), and a significant fraction of infections that are also asymptomatic [3] is in agreement with clinical evidence showing that HCWs are often exposed to SARS-CoV-2 also outside COVID-19 wards and may become infectious without symptoms [15–17]. Our finding that potentially infectious HCWs might be more common among younger HCWs suggests that it is important to vaccinate this young target population, if not yet vaccinated [18].

## Supporting information

**S1 File. Study protocol in original language.**
(PDF)

## Acknowledgments

We wish to thank participating care units for their collaboration in the realization of this study.

## Author Contributions

**Conceptualization:** Ville N. Pimenoff, Joakim Dillner.

**Formal analysis:** Ville N. Pimenoff, Miriam Elfström.

**Resources:** Ville N. Pimenoff, Kalle Conneryd Lundgren, Susanna Klevebro, Erik Melen, Joakim Dillner.

**Visualization:** Ville N. Pimenoff.

**Writing – original draft:** Ville N. Pimenoff.

**Writing – review & editing:** Ville N. Pimenoff, Miriam Elfström, Kalle Conneryd Lundgren, Susanna Klevebro, Erik Melen, Joakim Dillner.

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
