## [Decision Letter · Decision Letter 0]

29 Oct 2021

PONE-D-21-25144Potential SARS-CoV-2 infectiousness among asymptomatic healthcare workersPLOS ONE

Dear Dr. Pimenoff,

Thank you for submitting your manuscript to PLOS ONE. After careful consideration, we feel that it has merit but does not fully meet PLOS ONE’s publication criteria as it currently stands. Therefore, we invite you to submit a revised version of the manuscript that addresses the points raised during the review process.

Please make sure that the revised version will be prepared considering all the suggestions from reviewer 1, with a very careful consideration of the statistical aspects, as recommended.

We look forward to receiving your revised manuscript.

Kind regards,

Cristian Apetrei, MD, PhD

Academic Editor

PLOS ONE

Journal Requirements:

"This work was supported by the Karolinska University Hospital (KUH); the County Council of Stockholm; Erling-Persson family foundation; KTH Royal Institute of Technology; Creades and SciLifeLab. There were no roles for the funders in design or execution of the study, in analysis and interpretation of data or in the decision to submit for publication"

"This work was supported by the Karolinska University Hospital (KUH); the County Council of Stockholm; Erling-Persson family foundation; KTH Royal Institute of Technology; Creades and SciLifeLab. There were no roles for the funders in design or execution of the study, in analysis and interpretation of data or in the decision to submit for publication."

3. We note that you have stated that you will provide repository information for your data at acceptance. Should your manuscript be accepted for publication, we will hold it until you provide the relevant accession numbers or DOIs necessary to access your data. If you wish to make changes to your Data Availability statement, please describe these changes in your cover letter and we will update your Data Availability statement to reflect the information you provide

4. Your ethics statement should only appear in the Methods section of your manuscript. If your ethics statement is written in any section besides the Methods, please move it to the Methods section and delete it from any other section. Please ensure that your ethics statement is included in your manuscript, as the ethics statement entered into the online submission form will not be published alongside your manuscript

Reviewers' comments:

Reviewer's Responses to Questions

**Comments to the Author**

1. Is the manuscript technically sound, and do the data support the conclusions?

Reviewer #1: No

Reviewer #2: Yes

2. Has the statistical analysis been performed appropriately and rigorously? 

Reviewer #1: No

Reviewer #2: I Don't Know

3. Have the authors made all data underlying the findings in their manuscript fully available?

Reviewer #1: No

Reviewer #2: Yes

4. Is the manuscript presented in an intelligible fashion and written in standard English?

Reviewer #1: Yes

Reviewer #2: Yes

5. Review Comments to the Author

Reviewer #1: The primary objective of a secondary study was to quantify the proportion of healthcare workers who were potentially infectious. Their sick leave was classified as post-symptomatic or pre-symptomatic, and virus-positive subjects with no sick leave were considered asymptomatic. About 0.2% of he HCW at work were potentially infectious and pre-symptomatic (later had disease) and 0.17% were potentially infectious and asymptomatic (never took sick leave). In summary, 33% and 28% of all the 57 potentially infectious subjects were pre-symptomatic or asymptomatic, respectively.

Major revisions:

Include a statistical analysis section which lists and describes all the statistical methods used to analysis the data; cite the statistical software used.

Minor revisions:

1- Abstract: Replace “neither” with “either.”

2- Provide the percentage that corresponds to 16/57.

3- Table 2: Provide corresponding percentages for categorical factors. For instance, for sex provide the percentage corresponding to 44/(44+13) Female & 13/(44+13) Male, and 7488/(7488+1961) Female & 1961/(7488+1961) Male.

4- To assist in the review process, please add page and line numbers to the document.

Reviewer #2: This study identified the proportion of potentially infectious, nonsymptomatic HCWs which are either pre-symptomatic, asymptomatic or post-symptomatic. This topic is highlighted by WHO which mention the importance of this decisive factor on the spread of the pandemic.

The authors tested an impressive number of HCWs from Karolinska University hospital, using methods described in detail in references 7 and 8: serology and RT PCR. Previously published studies (references 6, 9 and 10) by the same team of researchers, support the authors' interest in the real-time pandemic situation in Stockholm.

The results of this study that the younger HCWs are potentially more infectious, based on real laboratory data, and the conclusion that they should get vaccinated, could be used like a model for other EU countries, to motivate their HCWs and general population to vaccinate themselves.

6. PLOS authors have the option to publish the peer review history of their article (what does this mean?). If published, this will include your full peer review and any attached files.

Reviewer #1: No

Reviewer #2: No

---

## [Author Response · Author response to Decision Letter 0]

2 Nov 2021

Following the Editors request and guidelines we have addressed the four points below, followed by our response to the reviewers comments.

1. We note that you have stated that you will provide repository information for your data at acceptance. Should your manuscript be accepted for publication, we will hold it until you provide the relevant accession numbers or DOIs necessary to access your data. If you wish to make changes to your Data Availability statement, please describe these changes in your cover letter and we will update your Data Availability statement to reflect the information you provide.

Response: The anonymized individual-level HCWs COVID-19 results (RT-PCR and serology) data (N=9449) with clinical features used in this study are deposited into b2share (https://b2share.eudat.eu/). The separate clinical dataset of 3981 HCWs who completed serology testing and a questionnaire about symptoms is only available in aggregated age-group format but made also available in the b2share portal. We will make the data files publicly available as soon as the paper is accepted for publication.

Response: Modified accordingly

Response: Revised and corrected one in press – article with the up-to-date citation information.

4. Have the authors made all data underlying the findings in their manuscript fully available?

Reviewer #1: No

Reviewer #2: Yes

Response: Please, see our response to the Q1 above.

Please, see below our on-by-one responses to the reviewers comments:

Reviewer #1: 

The primary objective of a secondary study was to quantify the proportion of healthcare workers who were potentially infectious. Their sick leave was classified as post-symptomatic or pre-symptomatic, and virus-positive subjects with no sick leave were considered asymptomatic. About 0.2% of he HCW at work were potentially infectious and pre-symptomatic (later had disease) and 0.17% were potentially infectious and asymptomatic (never took sick leave). In summary, 33% and 28% of all the 57 potentially infectious subjects were pre-symptomatic or asymptomatic, respectively.

Major revisions:

1. Include a statistical analysis section which lists and describes all the statistical methods used to analysis the data; cite the statistical software used.

Response: Statistical analysis paragraph was included into the methods section. Line 100.

Minor revisions:

2. Abstract: Replace “neither” with “either.”

Response: Modified accordingly. Line 31.

3. Provide the percentage that corresponds to 16/57.

Response: Modified accordingly. Line 109.

4. Table 2: Provide corresponding percentages for categorical factors. For instance, for sex provide the percentage corresponding to 44/(44+13) Female & 13/(44+13) Male, and 7488/(7488+1961) Female & 1961/(7488+1961) Male.

Response: Modified accordingly. Table 2.

5. To assist in the review process, please add page and line numbers to the document.

Response: Added accordingly.

Reviewer #2: 

This study identified the proportion of potentially infectious, nonsymptomatic HCWs which are either pre-symptomatic, asymptomatic or post-symptomatic. This topic is highlighted by WHO which mention the importance of this decisive factor on the spread of the pandemic. The authors tested an impressive number of HCWs from Karolinska University hospital, using methods described in detail in references 7 and 8: serology and RT PCR. Previously published studies (references 6, 9 and 10) by the same team of researchers, support the authors' interest in the real-time pandemic situation in Stockholm. The results of this study that the younger HCWs are potentially more infectious, based on real laboratory data, and the conclusion that they should get vaccinated, could be used like a model for other EU countries, to motivate their HCWs and general population to vaccinate themselves.

Response: We thank the reviewer for this important comment. We also believe that these results should motivate intervention policies for vaccinating the healthcare personnel at all ages in a systematic fashion.

---

## [Editor Report · Decision Letter 1]

10 Nov 2021

Potential SARS-CoV-2 infectiousness among asymptomatic healthcare workers

PONE-D-21-25144R1

Dear Dr. Pimenoff,

We’re pleased to inform you that your manuscript has been judged scientifically suitable for publication and will be formally accepted for publication once it meets all outstanding technical requirements.

Kind regards,

Cristian Apetrei, MD, PhD

Academic Editor

PLOS ONE
---

## [Editor Report · Acceptance letter]

9 Dec 2021

PONE-D-21-25144R1 

Potential SARS-CoV-2 infectiousness among asymptomatic healthcare workers 

Dear Dr. Pimenoff:

I'm pleased to inform you that your manuscript has been deemed suitable for publication in PLOS ONE. Congratulations! Your manuscript is now with our production department. 

Kind regards, 

on behalf of

Dr. Cristian Apetrei 

Academic Editor

PLOS ONE